# Human Footprints in the Karst Landscape: The Influence of Lime Production on the Landscape of the Northern Dalmatian Islands (Croatia)

Josip Faričić [1,*] and Kristijan Juran [2]

1 Department of Geography, University of Zadar, 23000 Zadar, Croatia
2 Department of History, University of Zadar, 23000 Zadar, Croatia; kjuran@unizd.hr
* Correspondence: jfaricic@unizd.hr

**Abstract:** Throughout history, the production of lime on the Croatian islands, which are mostly made of limestone and dolomite, has been an important economic activity. In the northern Dalmatian islands, which are centrally positioned on the northeastern Adriatic coast, lime was produced for local needs, but also for the purposes of construction in the nearby cities of Zadar and Šibenik. On the basis of research into various written and cartographic archival sources relating to spatial data, in addition to the results of field research, various traces of lime production have been found in the landscape of the northern Dalmatian islands. Indications of this activity in the insular karst are visible in anthropogenic forms of insular relief (lime kilns, small quarries, stone deposits) and in degraded forms of Mediterranean vegetation. This activity has also left its mark on the linguistic landscape in the form of toponyms, indicating that lime kilns were an important part of the cultural landscape.

**Keywords:** lime kiln; karst; insular landscape; environmental changes; northern Dalmatia; Croatia





## 1. Introduction

The object of the research that preceded this article is the impact of lime production on landscape changes in the northern Dalmatian islands. Lime was produced on the islands for local needs in the construction sector (as a binding material, and for the plastering and painting of inner walls) and in agriculture (to protect grape vines and fruit trees against pests and vermin). Lime produced on the islands was also used for the construction of sacred and secular buildings in Zadar and Šibenik [1–3].

Lime production consisted of the gathering of stones (limestone and dolomite fragments) rich in calcium carbonate and the construction of lime kilns in which a burning fire helped achieve high temperatures whereby the collected stones were transformed into a friable mass. Accordingly, there existed a widespread method of lime production as evidenced by archeological traces in various inhabited and economically used areas [4–15], in which basic raw materials (limestone and dolomite) and energents (fossil fuels and timber) were available [5,9,15,16]. In addition, lime production has been described in various historical sources (e.g., in specific chapters, the production of lime and its use were described by Marcus Porcius Cato in his work *De agri cultura*, 160 BC, and by Marcus Vitruvius Pollio in his work *De architectura*, 30–15 BC). High temperatures result in the thermal degradation of calcium carbonate ($CaCO_3$) into calcium oxide ($CaO$) in the form of friable matter (with the stones' preserved form), i.e., quicklime (when dolomite was used with calcium oxide, the dolomite lime contained magnesium oxide ($MgO$) and carbon dioxide ($CO_2$)) [15]. Quicklime was hydrated by mixing a small quantity of water with lime, whereby calcium hydroxide was obtained ($Ca(OH)_2$). Quicklime and hydrated lime were mixed with a large quantity of water, whereupon slaked lime was obtained, which in turn was used to produce lime milk.

The production of lime was accompanied by the exploitation of surface layers of carbonate sediment rock and tree-felling, which directly resulted in changes to relief

structures and the degradation of vegetation [5,17]. This degradation further caused the washing away of prevailing shallow soils on insular slopes and the exposure of rock to various slope-related processes. Simultaneously, the felling of trees and shrub-like plants decreased biodiversity, and their use for heating released $CO_2$ into the atmosphere [15].

Lime production simultaneously influenced and caused substantial long-lasting changes in the environment and the landscape. Thus, it is clearly a form of natural resource usage that differentiates the Anthropocene as the most recent period in the geological history of the Earth [18–22]. In this manner, lime kilns can be considered to be techno fossils; that is, unique material remnants of the technosphere [23].

In recent years, archaeological excavations of lime kilns have been carried out systematically in Italy [6,7], Greece [11], Spain [9], France [12,13], and Switzerland [10]. To date, with rare exceptions, such as Zlatunić's study of lime kilns at the Dragonera site in Istria [24], no such research has been conducted in Croatia. Thus, we do not have valuable archaeological data that could be compared with other sources. However, by comparing the diverse available and investigated sources of spatial data, a sufficiently meaningful picture has been formed that indicates the influence of lime production on the formation of the insular landscape in the northern part of Dalmatia. To the best of our knowledge, this is the first historical-geographical presentation of lime production covering a specific part of Croatia. We hope that it will encourage more systematic research within various scientific fields (geology, geography, archeology, etc.).

## 2. Study Site

The northeastern Dalmatian islands are a fragmented and spatially scattered archipelago situated off the coast of Northern Dalmatia, which is the central region of the eastern shore of the Adriatic Sea (Figure 1). They are situated within the gravitational reach of two prominent Croatian coastal centers—Zadar and Šibenik—which means they have the closest social and economic ties with these two cities [25].

The northern Dalmatian islands mostly consist of Cretaceous and Eocene limestone and dolomite rocks [26–30], which in many places, especially on steeper slopes and in the immediate vicinity of the sea, are either not covered with soil or lie beneath a very loose, thin layer. Most of the Zadar and Šibenik islands are made of Upper Cretaceous ore-bearing limestone (Cenomanian–Maastrichtian, $K_2^{1-6}$) (Figure 2). The thickness of their strata varies, ranging from thin plates to thick layers (1–2 m), and there are also massive strata, the majority of which are between 30 and 80 cm thick [29].

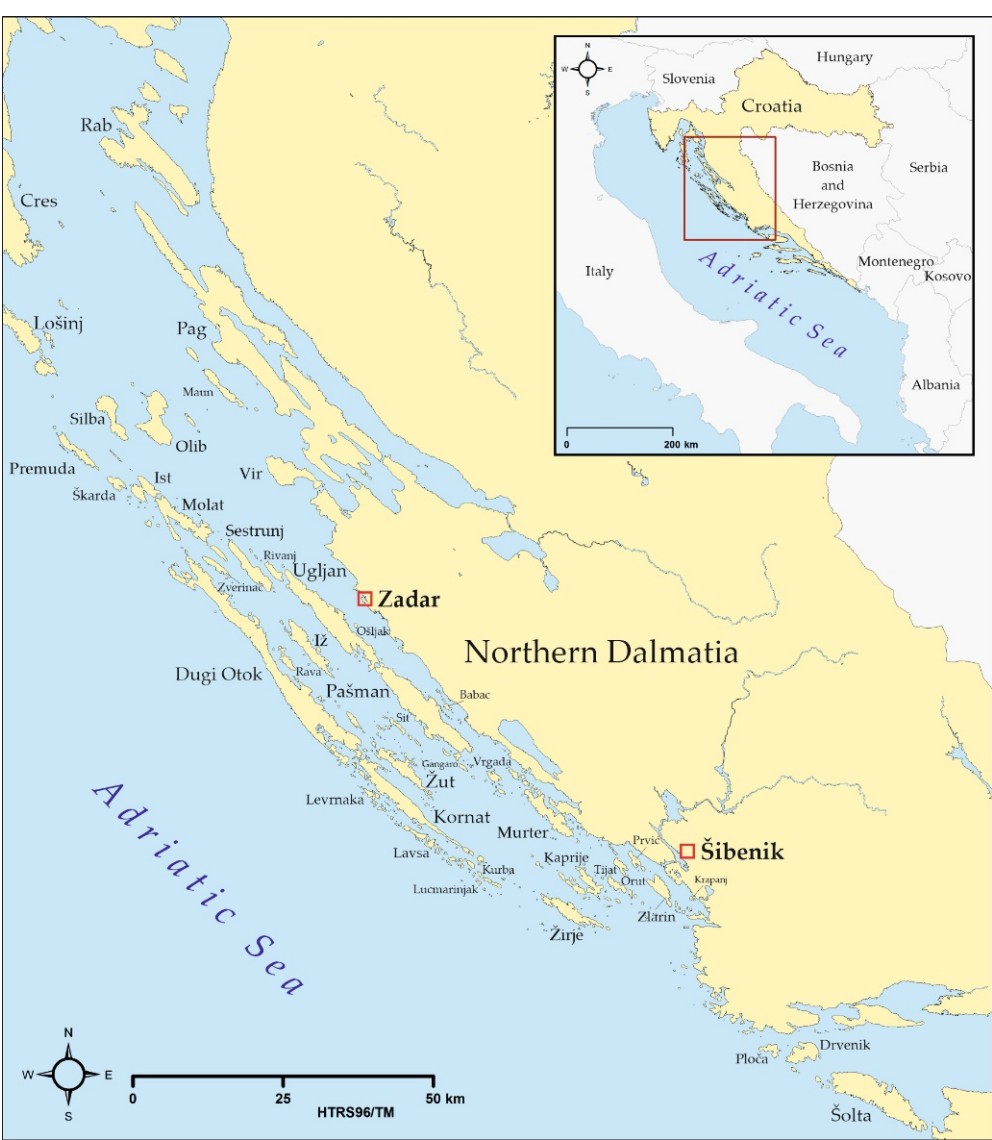

**Figure 1.** The geographical location of the northern Dalmatian islands.

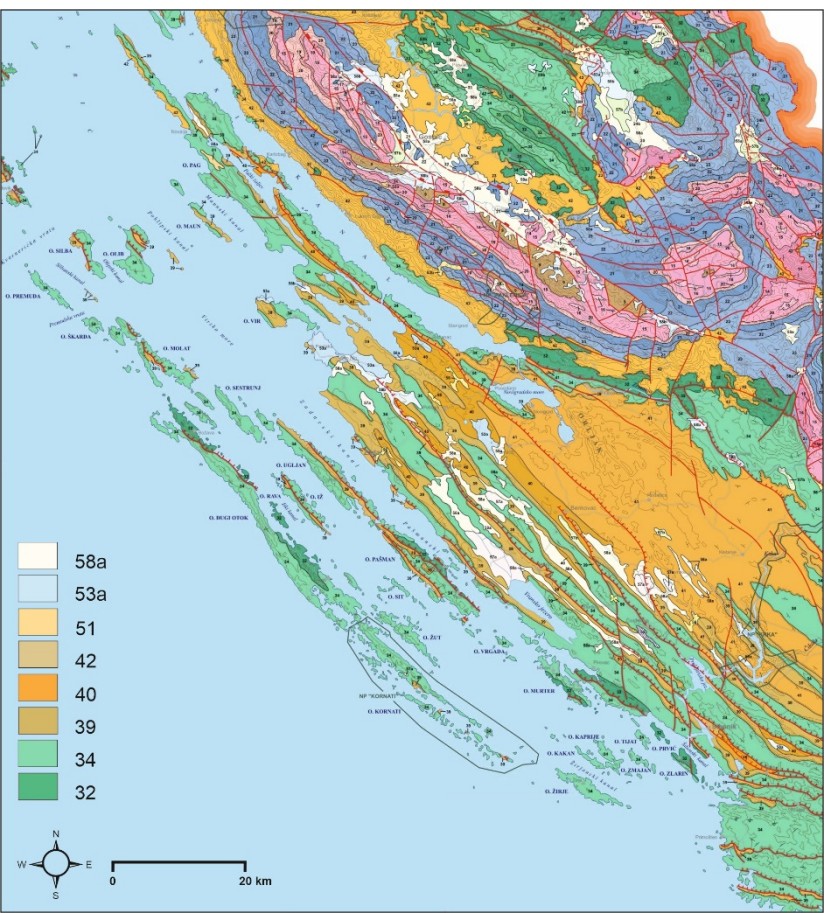

**Figure 2.** Excerpt from Geological Map of the Republic of Croatia, original scale 1:300,000, depicting the northern Dalmatian islands (explanation of numbers according to the map's legend: 32—Lower Cretaceous dolomite, 34—Upper Cretaceous ore-bearing limestone, 39—Eocene Foraminifera limestone, 40—Eocene flysch deposits, 42—Paleogene-Neogene Limestone breccia, 51—Neogene deposits, 53a—Pleistocene fluvial deposits, 58a—Holocene diluvial and proluvial deposits).

## 3. Materials and Methods

The study was based on repeated field observations, the results of toponomastic explorations, and the analysis of written and cartographic archival sources.

During a field trip to the northern Dalmatian islands, we identified many sites with remnants of lime kilns, and in conversations with the local population, we elicited numerous toponyms that name the micro-locations where lime was produced. This toponymic data was supplemented with data from field research conducted by the Center for Adriatic Onomastic Research at the University of Zadar. Through field research of the North Dalmatian islands, completely new data were collected in relation to those on the available official topographic maps at a scale of 1:25,000, in addition to those on modern cadastral maps. Because the remains of lime kilns are not shown on these maps, given the scale of the maps and the cost-effectiveness of printing toponyms, only one toponym is written on them that directly refers to the production of lime (Cape Japleniško on the island of Ugljan).

During our archival research, we utilized original medieval and early modern documents and maps kept on file at the State Archives in Zadar and the State Archives in Šibenik. The original documents were first translated from Latin and Venetian Italian, and then analyzed in the context of socio-economic activities on the islands in the area of Zadar and Šibenik. The analysis of cartographic sources of spatial data was based on the search for confirming evidence about the production of lime on more than one hundred early modern topographic maps and cadastral plans that were drawn up in local surveying-mapmaking

workshops in Zadar. On several maps, drawings of lime kilns were found, in addition to toponyms that indicated locations at which lime was produced.

Considering that lime production on the northern Dalmatian islands came to an end in the late nineteenth century, no detailed descriptions of the production process have survived, nor have any of the kiln structures survived; rather, there are only remnants of burnt rocks at their former locations. For this reason, a published ethnographic description of lime production on the island of Šolta was extremely useful, because this island is situated in the immediate vicinity of the islands examined in the current study, and the production of lime in the traditional manner continued there until the mid-twentieth century. This is documented by photographs showing lime kilns on Šolta, which are similar in construction to those depicted on the old maps and described in archival documents.

## 4. Results

For centuries, the insular landscape was formed by various human activities. The basic landscape patterns are similar to those which are documented throughout the broader Dinaric region, which is marked by karst as the basic hydrogeological-geomorphological phenomenon [31–37]. Anthropogenic influence was manifested in the reshaping of relief forms whose geological basis consists of carbonate rocks and, in rare zones, clastic sediments and soils developed on them on terrains with lesser slopes. Important among these activities was the exploitation of stone used for the construction of residential and economic facilities, for the terracing of slopes (the building of drystone walls perpendicular to the direction of water runoff and slope processes to prevent erosion and the washing away of soil), and the marking of boundaries between properties, in addition to the building of various elements of traffic infrastructure (ways and roads, bridges, ports and smaller harbors), and for the production of lime. For local housebuilding and various other needs of the coastal towns (the construction of port wharves, street paving, etc.), stone has been used since ancient times. Such stone was extracted from quarries on the islands of Sestrunj [36,37], Dugi otok [37,38], Lavdara [39], Žut [40], Kornat [41], and Pag [42,43], on all the major islands in the local waters of Šibenik [1], and on many islets in the northern Dalmatian archipelago. This is evidenced by many material remains, primarily quarries.

The shaping of the insular landscape was greatly influenced by the felling of trees for firewood, housebuilding, shipbuilding, limestone burning, and clearing to obtain arable land and pastures. Given that agriculture, along with seafaring, was the main economic activity in this predominantly rural area, the landscape was dominated by terraced slopes on which olive groves and vineyards were built; a large part of the island area consisted of pastures, while wooded areas gradually subsided [44].

In accordance with the deagrarianization and even more intense depopulation that occurred in the mid-twentieth century, many olive groves and vineyards were abandoned, and due to the succession of native vegetation and the spread of allochthonous species such as the Aleppo pine, older landscape patterns were covered or completely replaced by new ones: the old drystone walls are no longer even visible and are falling apart, and pine forests and maquis have spread into many areas where olive groves and vineyards used to be. This is clearly indicated by bitemporal pairs of photographs taken at the beginning of the twentieth and twenty-first centuries (Figure 3), in addition to bitemporal pairs of aerial images and a comparison of cadastral data on land use categories in the middle of the 20th century and at the beginning of the 21st century [44].

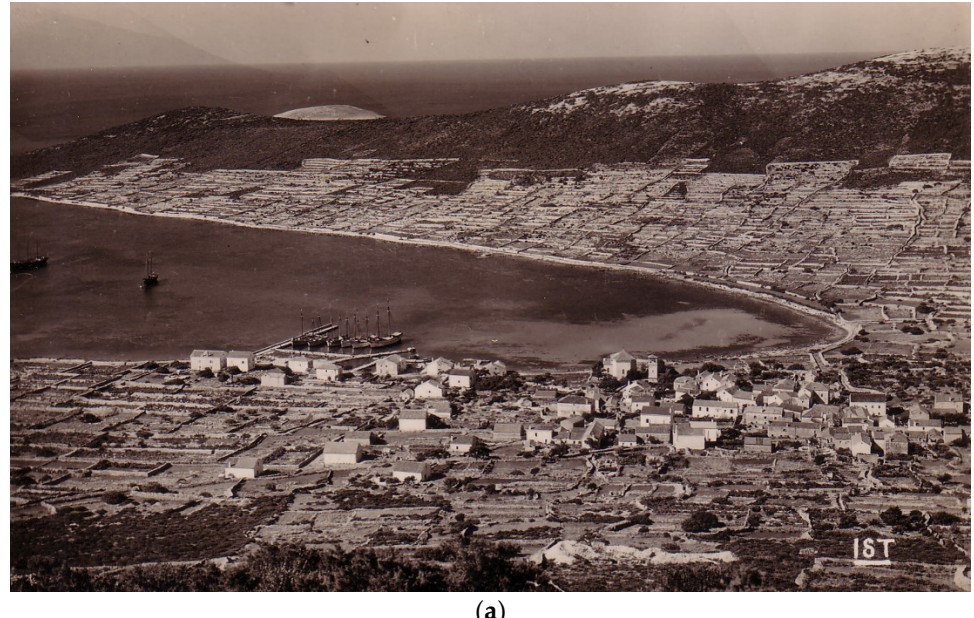

(**a**)

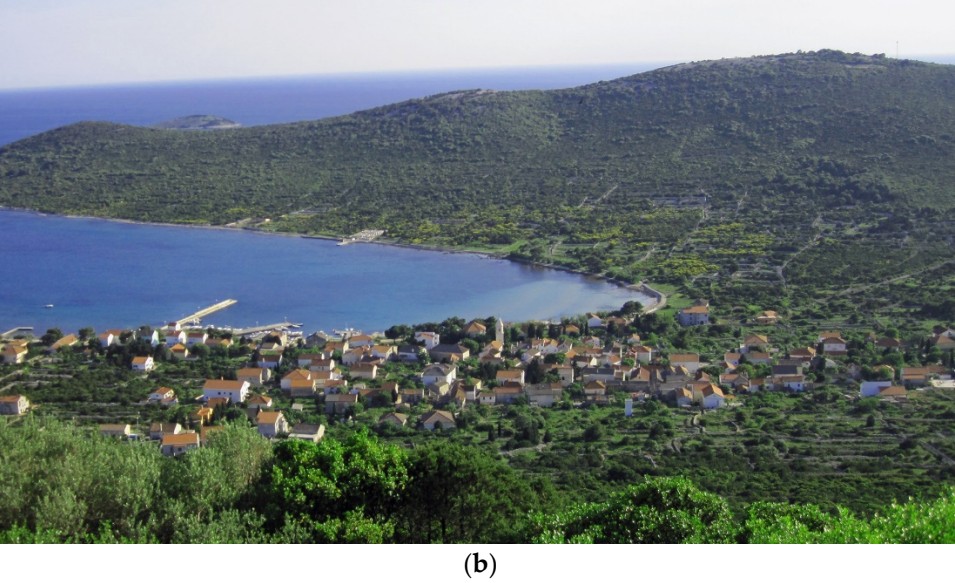

(**b**)

**Figure 3.** The changing insular landscape as documented on the island of Ist: (**a**) the island's landscape in the early 20th century (family collection of Zvonimir Šuljak); (**b**) the landscape photographed in 2010.

In addition, as tourism developed, only the narrow coastal belt in the zones of settlements or the most attractive island bays were utilized more intensively. Drystone walls and stone-constructed elements of the port infrastructure were replaced by new structures in which concrete predominates, and the old stone centers of each settlement are surrounded or completely replaced by concrete, glass, and aluplast structural elements of holiday cottages, apartments, and restaurants. The scale of this concretization is significant. According to population and dwelling tallies from the 1971 and 2001 censuses, the total housing stock in the northern Dalmatian islands increased from 11,291 dwellings with a total area of 785,656 m$^2$ to 37,095 dwellings with a total area of 2,766,918 m$^2$, with holiday dwellings (cottages and apartments) accounting for 17.8% of the total number of dwellings in 1971, and holiday apartments accounting for 53.3% of the total number of dwellings in 2001. In the same period, the number of permanent residents on the northern Dalmatian islands decreased from 39,872 to 30,703 [45–47].

### 4.1. Locational Factors of Lime Production

There are several key locational factors that influenced lime production. Lime kilns were built in the immediate vicinity of the main raw materials (limestone and dolomite) and energents (wood), and also in places suitable for the transport of lime. This procedure was used on parts of the coast where the sea was sufficiently deep and easily accessible and, where needed, their accessibility was increased by the construction of stone wharves [48].

Most of the lime kilns on the islands were located in the area with Upper Cretaceous ore-bearing limestone (Cenomanian–Maastrichtian, $K_2^{1-6}$), which was easy to break and use to build limestone kilns. Eocene Foraminifera limestone ($E_{1,2}$), which is found on the islands of Silba, Olib, Pag, Vir, Molat, Ugljan, Pašman, Iž, and Kornat [30], was also exploited for lime production.

The smallest limestone kiln sites were found in the area dominated by Lower Cretaceous dolomite ($K_1$), which is mostly massive and bulky [28]. Although this form of dolomite is found on the islands of Dugi Otok, Pašman, Murter, Iž, Prvić, Zlarin, and Rava, lime production from dolomite fragments was recorded only on Rava.

In areas with Eocene flysch deposits ($E_{2,3}$), Paleogene-Neogene limestone breccia (Pg, Ng), Neogene deposits ($M_3$–$M_5$) and Holocene diluvial and proluvial deposits ($dprQ_2$) on the island of Pag and on part of the island of Vir with Pleistocene fluvial deposits ($a$-$aQ_1$), no lime kiln sites have been recorded. This also confirms that lime kilns were located only in the immediate vicinity of the available raw material in order to facilitate the movement of the limestone to the kiln, so that the distance from quarry to kiln did not exceed several meters. The lime kilns were also built in the immediate vicinity of the shore, so that the lime produced there could be easily loaded onto boats for further transport (Figure 4).

The geological setting also influenced the types of kilns that were built. Although archeological excavations in Italy [5], Spain [9], and France [13] identified several types of lime kilns, including those buried in the ground, the lack of soil or clastic material in which a sufficiently deep hole could be easily excavated meant that the kilns on the northern Dalmatian islands were built above ground with a dome on rock outcroppings along the coast.

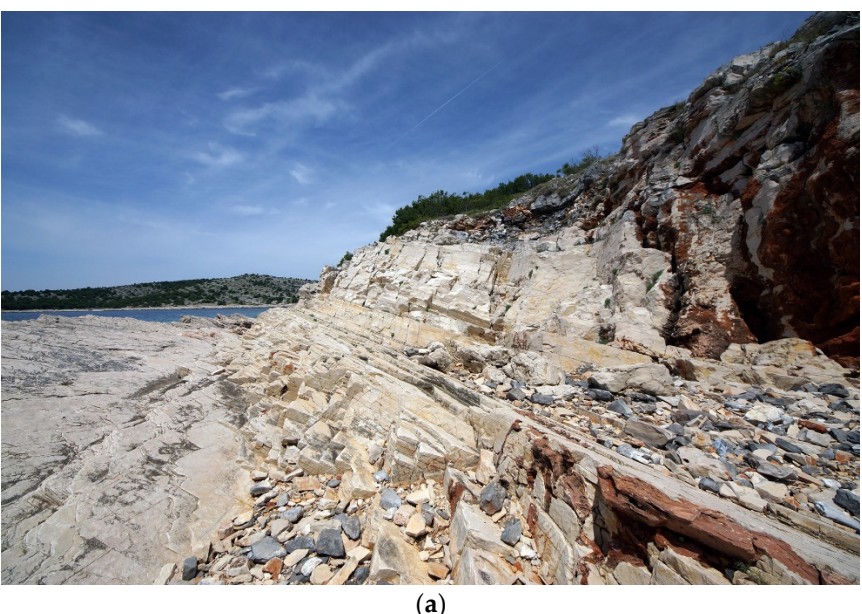

(**a**)

**Figure 4.** *Cont.*

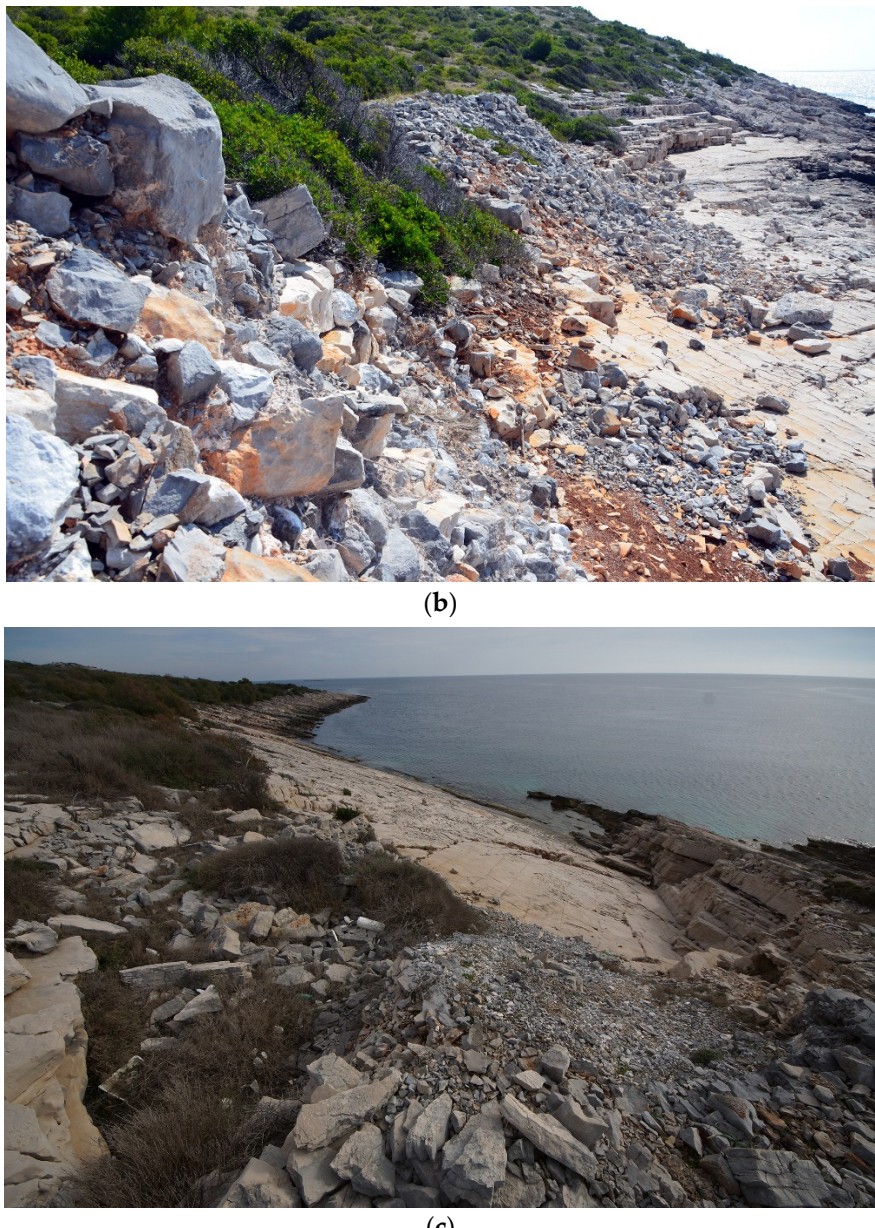

**Figure 4.** Remnants of lime kilns on three islands in northern Dalmatian (photo by J. Kale): (**a**) Orut, (**b**) Levrnaka, and (**c**) Žirje. The photographs depict only the remains of burnt stones and rubble at the extraction site of stones that had been arranged in the form of a kiln nearby, because the actual lime kilns have not been preserved.

To provide sufficiently high temperatures in lime kilns over several days (ca. 1000 °C), needed for thermal degradation of calcium carbonate (and magnesium carbonate when dolomite was used), it was necessary to use a substantial quantity of firewood. This increased the intensity of tree and shrub felling.

It is difficult to quantify the influence of lime production on local plant life because there are no precise relevant data. For example, Sarabia et al. [17] noted the impact of lime production on deforestation based on a study of landscape changes in Dren in the Trentino region of northern Italy. However, they did not specify the number of trees cut down because they did not have sufficient reliable data. They only pointed out that, for many lime kilns, an *enormous quantity of wood* was used [17]. Based on relatively recent archival documents relating to lime production in Liguria in the nineteenth century, Vecchiattini estimated that for the production of 1 quintal (100 kg) of lime, 1 quintal of wood was

needed to heat the lime kilns [7]. Because the total amount of lime produced is unknown, Vecchiattini could not determine the total amount of wood burned in the lime kilns.

In the sources available in the archives, the data relating to the Zadar and Šibenik islands are not consistent; they are not chronologically continuous or geographically solid, and they do not contain statistical information that might indicate the scope of lime production. It is only possible to surmise that the impact was significant, based on the data collected from people who were professionally active in the production of lime towards the mid-twentieth century. In this regard, the best documented lime production is that of the island of Šolta, situated in central Dalmatia, which is connected to the area directly bordering on the researched area of the northern Dalmatian islands. Blagaić and Burica [49], who spoke with old lime producers on Šolta, documented the fact that the production of 10 tons of lime required 200 loads of macchia (holm oak, strawberry-tree, mastic tree, etc.) and from 3 m$^3$ to 5 m$^3$ of pine wood (or any other wood). Pine wood was used for the initial stoking of fire in lime kilns, whereafter, when it burned off, the fire stoking would continue with bundles of dried macchia. Each macchia load consisted of branches tied together in a bundle-like shape. It was possible to grasp these macchia bundles and load them onto the heads or backs of people who carried them from the place where wood was felled to the location where lime was produced. The size of such a load amounted to approximately 1 fathom (depending on the person who made the macchia bundles, ca. 1.7 m on average). This was a rudimentary chore that did not offer the possibility of using a precise system of measurement; regardless, no such system was necessary (throughout many centuries on Croatian islands under Venetian rule, *passo Veneziano*, "the Venetian fathom" (equivalent to 1.73823 m) was used) [50].

The lime kilns on Šolta Island had varying lime production capacities, mostly ranging from 5 to 20 t [49]. A particular lime kiln would produce lime several times during the course of a year, depending on the needs of the local population, orders made by other people, and the annual schedule of agricultural and fishing activities. There were approximately 600 lime kilns on Šolta Island. However, their production capacities remain unknown, and there is no indication as to whether they were used every year and how many times they were used in a given year. The size of these lime kilns can be estimated on the basis of photographs taken in the mid-twentieth century (Figure 5) and published in Šule's photo-monograph on Šolta Island [51]. Remnants of many such kilns on that island were also found during a study conducted by a team of geographers and geologists within the framework of the scientific project titled "Geographical Basis of Small Croatian Islands Development," University of Zadar (Šolta, summer 2005). Judging by a comparison of lime kiln sizes on Šolta Island and on the islands of northern Dalmatia, those in the latter area were smaller, which means their individual single capacity of produced lime would have been up to 5 t, with the quantity of macchia used being approximately 100 loads and firewood between 2 m$^3$ and 4 m$^3$.

Although there are partial material remnants of lime kilns on many islands, none have survived in a form that would enable a somewhat accurate reconstruction. Moreover, it is still unknown how many of these would have been functional at any one period of time, and we do not know how many times a year lime was produced there. Moreover, given that trees (holm oak, pine, cypress) and shrub-like plants (strawberry-tree, myrtle, mastic tree, laurestine, mock privet, etc.) were used for several different activities (tree felling for the purposes of housebuilding, naval architecture, firewood, tool making, fire stoking during fishing for small pelagic fish at night, etc.), or were removed because of vineyard or olive grove construction and for obtaining more grazing pastures, it is not possible to define the exact share of plant mass felled and used in lime production. It is only possible to determine that we are dealing with an unknown but significant quantity that surely had an impact on vegetation degradation, consequently resulting in intensified soil erosion.

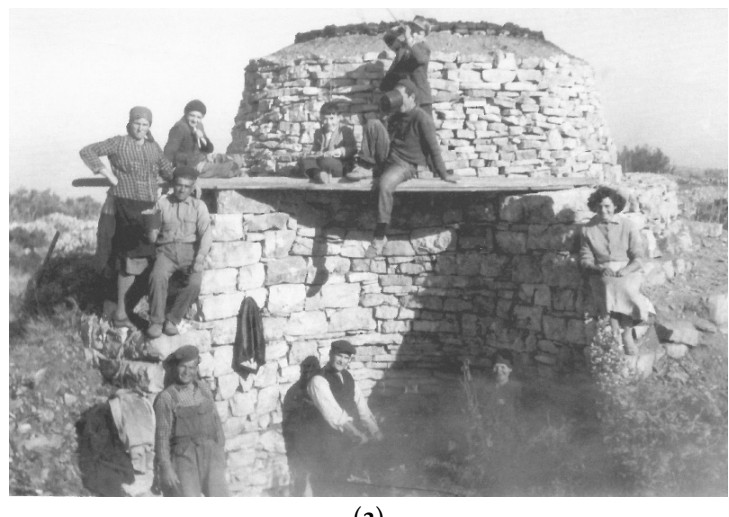

(**a**)

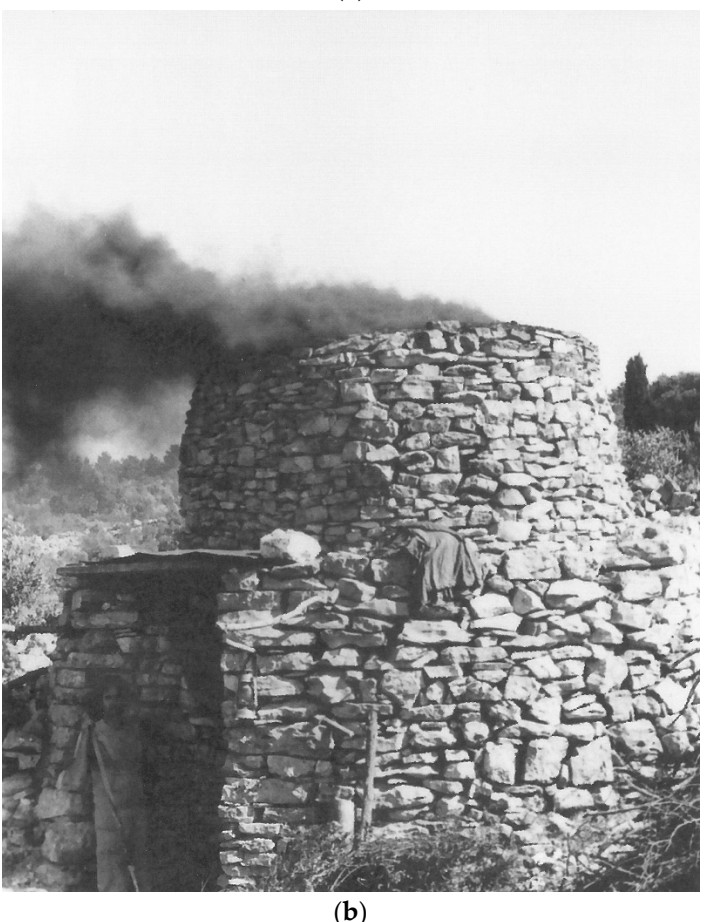

(**b**)

**Figure 5.** Lime kilns on Šolta Island from mid-20th century: (**a**) a kiln under construction; (**b**) a kiln in which a fire has been lit.

The proportions of lime production are approximately indicated by the population data of the northern Dalmatian islands. The first demographic data stem from the early sixteenth century, by which point the network of settlements had taken a definitive shape, whose fundamental features have not changed to the present day. However, the demographic data is not fully comprehensive: until the mid-nineteenth century, the data is not consistent nor does it comprise all of the islands simultaneously. In the year 1527, the Zadar islands under immediate administration of the city of Zadar had a population of

6702 [52,53], and in 1608 these islands had 5873 inhabitants [54]. In the year 1759, the Zadar islands had a population of 10,498, and 11,806 people lived on them in 1818. These censuses did not comprise the islands of Vir (which belonged to the commune of Nin) or the island of Pag, which was divided between the Commune of Pag and the Commune of Rab [44]. In 1802, the island of Vir had a population of 250 inhabitants [55], whereas 3162 inhabitants were recorded as living on the island of Pag in 1813 [56]. The islands in the local waters of Šibenik had a population of 12,484 inhabitants in 1585, and 8630 inhabitants were living there in 1599 [57,58]. In the eighteenth century, no comprehensive censuses were taken on the islands located in the Commune of Šibenik. All northern Dalmatian islands were included in the same census in 1830. In that year, all of Zadar's islands had a total population of 16,083 inhabitants, whereas all of the islands in the vicinity of Šibenik had a population of 7624 inhabitants [59]. Regardless of the fact that the statistical data was faulty until the first consistent census was carried out by the state authorities in 1857 [60], it is possible to detect a gradual increase in population (Table 1), which means that the number of houses gradually increased. Consequently, lime was an indispensable construction material in that period. Demographic growth intensified from the mid-nineteenth century until the first decades of the twentieth century, so one can justifiably assume that this was accompanied by an increase in the production and local consumption of lime.

**Table 1.** Population growth in the northern Dalmatian islands from 1857 to 1948 [60].

| Year | 1857 | 1869 | 1880 | 1890 | 1900 | 1910 | 1921 | 1931 | 1948 |
|---|---|---|---|---|---|---|---|---|---|
| Population | 27,914 | 30,211 | 33,464 | 37,045 | 42,024 | 45,124 | 49,661 | 48,400 | 48,421 |

In addition, on the northern Dalmatian islands, a significant quantity of lime was used in vineyards for the protection of grape vines. In the last decades of the nineteenth century, viticulture on these islands experienced an economic upturn when vineyards in western and southern European countries were affected and infested by the grape-vine pest phylloxera [61]. This favored the placement of Dalmatian wine on the European market until the moment when phylloxera also infested Dalmatian vineyards. Due to the increased demand for Dalmatian wine and an increase in the population of the islands in the local waters of Zadar and Šibenik, much agricultural land was converted into vineyards. Previously, these converted areas had been mainly used as olive groves, orchards, and pasture land [62]. In the year 1900, in the northern Dalmatian islands, an area of 8321 hectares was used for vineyards, of a total of area of all cadastral municipalities amounting to 90,188 hectares [63]. The grape vines in these vineyards were treated with the so-called Bordeaux mixture, containing blue vitriol and lime, to prevent downy mildew, a disease that frequently afflicted vine leaves. However, although downy mildew could be prevented by pouring the Bordeaux mixture over the grape vines, phylloxera was an incurable disease. In the last decade of the nineteenth century, this disease appeared in Dalmatia, and in the first two decades of the twentieth century it brutally ravaged a large number of Dalmatian vineyards. With agrarian overpopulation, the abovementioned circumstances caused an economic crisis that resulted in intense emigration not only from the islands to the cities on the mainland, but also to foreign countries, especially the United States, various countries in South America, and Australia. This resulted in a decrease in the population of the northern Dalmatian islands and a significant decrease in viticulture [62]. At the same time, the islands were influenced by modernization, which had an impact on the building of houses and led to the extinction of traditional crafts. Lime production on the islands was abandoned, which meant that those who needed lime had to purchase it in specialized construction material stores in nearby towns. A similar situation has been observed in other Mediterranean areas, where the traditional production of lime was replaced by the industrial production of this building material. For example, in Andalusia, Spain, there is a movement to protect traditional lime production as an intangible cultural heritage [8],

with the indication that there is clear interest in the registration, protection, and thus preservation of this traditional craft, regardless of its negative impact on the environment.

*4.2. The Oldest Archival Confirmation of Insular Lime Production*

The traditional craft of lime production was practiced on almost all of the northern Dalmatian islands. This fact is corroborated by numerous archival documents dating to medieval times. It was recorded that Zadar noblemen towards the end of the thirteenth century bought lime from Šibenik traders, which directly helps us conclude that, even then, lime production activity was highly developed on the islands located in the local waters of Šibenik [3]. A similar conclusion could be applied to the islands in the local waters of Zadar, which is, for example, suggested by data from the fourteenth century regarding lime kilns on the Kornati islands [64]. Given that there existed a continuous need for lime, not only in the private use, but also in the public construction sector, communal authorities strived to monitor its production. Accordingly, in 1381 the Šibenik Council of Nobles banned lime exportation without prior consent of the Duke. At the same time, there were incentives for the construction of lime kilns, which resulted in an increase in lime production on the islands in the coastal waters of Šibenik throughout the next century. This production itinerary was significantly influenced by substantial public and private construction works in the urban fabric of the Šibenik area (construction of the cathedral, the cistern, and palaces of the nobility, in addition to rampart fortification), but also by the necessity to fortify rural estates due to the growing danger of Ottoman invasion. Lime was also sought after for the construction of summer houses on the islands of Prvić and Zlarin, and demand for that material would eventually reach its peak in the first half of the sixteenth century, when the monumental fortress of St. Nicholas was built [3]. Therefore, it is not surprising that the most comprehensive data on lime production on the islands in the coastal waters of Šibenik originate from the period between 1450 and 1550. Within that timeframe, the existence of at least 30 contracts for the construction of insular lime kilns was recorded, whereby the island of Žirje was the biggest construction hub, followed by the islands of Kaprije, Orut, and Tijat [3,65]. It is clear that these four islands, with Žirje at the forefront, were continuously, for many centuries, the main centers of lime production in the Šibenik region. In other words, what Šolta was for central Dalmatia, Žirje was for the Šibenik section of Northern Dalmatia. The masters from Šolta periodically carried out their activities in the Šibenik area. For example, in 1421, Filip Ivanov and five other craftsmen from Šolta constructed a lime kiln on the island of Tijat [66]. It is well known from the archival news that significant shipments occurred in 1539 of lime from Šolta for the construction of the Fortress of St. Nicholas near Šibenik [67]. According to Šibenik sources, lime kilns were built by master craftsmen who were lime producers (*magistri calcariarum*) or masons (*murarii*). This activity included, in addition to the construction of lime kilns, the acquisition of wood, the firing of kilns, and the transporting of lime by sea to its destination. The outlays incurred for lime acquisition and firing a kiln accounted for up to one-third of the overall expenses. The documents quite often indicate the quantity of wood needed or used, which gives additional value to the data dating from 1451 regarding the planned acquisition of 2000 loads for the firing of one lime kiln on Žirje. The width of lime kilns built in the observed period (1450 to 1550) on the islands in the coastal waters of Šibenik ranged from 10 to 20 cubits, and was most frequently 13 cubits (inner diameter) or approximately 8 m in today's units of measure, and the height was related to the width. The kilns often had two fire doors and two vaults (*cum duabus portis et duobus voltis intus*) [3].

Late medieval lime production in Zadar was less studied than that in Šibenik, but the aforementioned parameters regarding the socio-historical context and conditioning of lime production on the islands in the local waters of Šibenik could easily and without significant alterations be applied to the insular area of Zadar. Because Zadar was an administrative and military center in the heart of Venetian Dalmatia, the demand for lime in that city was to a certain extent greater than in the neighboring commune. In addition, the islands in the coastal waters of Zadar were significantly richer in wood and shrubs than the islands in

the Šibenik region. Therefore, it is not surprising that the records suggest that ten farmers from the Šibenik district were contracted to gather wood for the construction of a lime kiln for two Zadar noblemen on the island of Ugljan [68]. The importance of lime kilns and the apparent richness of wooden construction material on that island, especially in the region of Kukljica, Kali, and Lukoran, were demonstrated by court proceedings initiated from the sixteenth to the eighteenth century by the monks of the Monastery of St. Dominic against laborers from the said region. These proceedings entailed litigation in relation to insular woods and pastures. From the depositions given in court in 1600, it can be concluded that many people collected wood unhindered in the forests near Kukljica and constructed lime kilns. During the war for Cyprus (1570–1573), two lime kilns were constructed by the noblemen Detrico and Cedolini, and after the war two or three (depositions do not concur in this regard) were constructed by the Srakić brothers of Kali, whereas Andrija Benić of Kukljica constructed only one in 1596. These lime kilns were usually fired with wood from the forests of Kukljica (*boschi di Cuclizza*). Some lime kilns on Ugljan were recorded in the archives as important landmarks. According to a document from 1650, the southeastern limits of the Preko settlement extended from the Brgačielj hill to the demolished lime kiln named Murandovac, located near the coast. Two years later, local islanders stated that all the other islanders were free to use the forests of Ugljan extending from Prkljug to Japnjenica. This was, of course, contested by the landowning elite [69]. If we consider the islands in the coastal waters of Zadar, not only Ugljan, but also the island of Pašman was an important center of lime production [70]. This fact was mostly illustrated by terminations and orders made by Dalmatian general proveditors in relation to the regulation and organization of public works and the construction of public facilities. For example, in 1661, a proveditor instructed that the construction of lime kilns be completed in the villages from Vrgada to Ugljan, among which, in terms of their spatial logic, are also all settlements on the island of Pašman [71]. Because it was necessary to ensure other raw materials and products for public needs, the proveditor's termination of 1705 defined specific labor obligations of certain insular village settlements as follows: the inhabitants of eight settlements on Pašman and the neighboring island of Vrgada were obligated to build lime kilns and transport lime to Zadar; the islanders of Ugljan had to transport sand, soil (*terra rossa*), and stone (*pietre*) to Zadar; the islanders of Dugi Otok transported stone slabs (*pianche*), refined stone, and sometimes coal; whereas the inhabitants of islands of Sestrunj, Molat, Premuda, and Vir only had to transport coal [72]. The well-preserved documentation of the general proveditor Vincenzo Vendramini (1708–1711) shows that the most significant share of lime for public works in Zadar was produced on the island of Pašman; that is, almost all Vendramini's terminations and orders regarding lime production (in Dalmatia) refer to Pašman and Šolta. In this three-year period, at least five lime kilns were built on Pašman. Half of the lime produced had to be transported to Zadar in the islanders' boats and subsequently handed over to the people in charge of public works, whereas the other half was kept by islanders for their own specific needs [73].

*4.3. Cartographic Confirmations of Lime Production in the Northern Dalmatian Islands*

Old maps are an important but not always reliable source of spatial data. Their authenticity depended on the methods of collecting the relevant data, which were subsequently processed and graphically visualized for different purposes under the influence of various geographical and thought perceptions and perspectives. Until the beginning of the nineteenth century, when the first consistent geodesic survey was taken in the Croatian insular area, more detailed maps of spatial entities were made based on topographic observations and simple measuring procedures. Therefore, the system of cartographic signs was not standardized nor did it consistently encompass larger spaces. There was no chronological continuity in the advancement of quality of mapping in relation to the same space, so most often it was not possible to reconstruct changes in a landscape that was the theme of depiction. Despite these shortcomings, many maps contain precious information relating

to various economic activities, such as lime production. The depictions on maps often correspond to the data in archival documents.

The oldest depiction with explicit mention of lime production is a map of the Zadar and Šibenik region (*il vero ritratto di Zarra et di Sebenico co diligenza ridotta in questa forma*) which was made in Venice in 1570 by the Šibenik-based Martin Rota Kolunić. Thereafter followed reproductions of the aforementioned map made by many others, mostly Venetian cartographers (for example Giovanni Francesco Camocio on the map *Sebenico et contado citta nella dalmatia confina[n]te con Zara d[el]li Ill[ustrissi]mi S[igno]ri Veneciani al p[rese]nte da Turchi molestado, published in the isolario Isole famose porti, fortezze, e terre maritime sottoposte alla Ser.ma Sig.ria di Venetia ad altri principi Christiani, et al. Sig. or Turco, nouamente poste in luce* in Venice in 1571). In Kolunić's original map kept at the National library in Paris [74], on the drawing of Žirje there is text that reads *Qui si fa la Calcina* ('Lime is produced here', Figure 6). The same map depicts all the other islands in the coastal waters of Šibenik, but apart from their geographical names there are no other specific notes. This means that Kolunić, an expert in the geography of the Šibenik region, had a reason to note Žirje as the main lime production area.

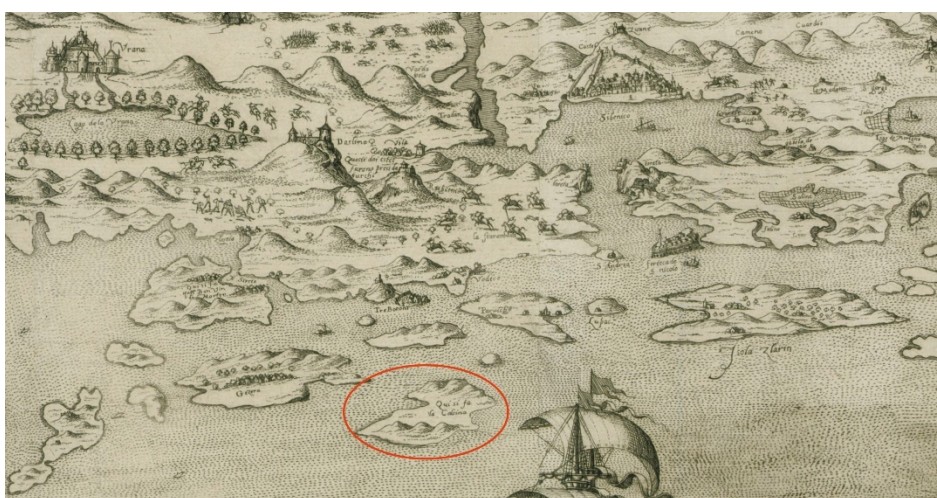

**Figure 6.** Excerpt from Kolunić's map of the Šibenik region, Venice, 1570. Red ellipse denotes Žirje Island entitled *Qui si fa la Calcina* (Lime is produced here).

During the seventeenth and eighteenth centuries, numerous hand-written cadastral-topographic large-scale maps were made. These were made by cartographers in the service of the Venetian administration in Dalmatia, whose seat was located in Zadar. The basic purpose of these maps was to record land properties, especially in cases when they were the object of purchase and gift contracts, or in cases when there was litigation in relation to land ownership. The making of such maps was preceded by uncomplicated surveying activities that encompassed only the land property that was the subject-matter of court or notarial proceedings, but not the entire area under Venetian administration. These activities resulted in achievements of varying quality that primarily depended on the personal competencies of a particular surveyor-cartographer. However, regardless of the differences in approach and scope of the workmanship and quality of graphic visualization of geographical reality, surveyors regularly chartered all relevant geographic facilities that had any economic value. Such facilities in insular space would invariably include lime kilns. Among these cartographic depictions, three were singled out that, each in its own way, indicate a certain particularity in comparison with similar but simpler content-related depictions.

A property sketch at the confines between Kali and Kukljica on Ugljan Island (Figure 7), along the banks of Mala Lamjana cove (*Porto di Lamgnane*), was made in 1736 [75]. It contained drawings of two lime kilns (*Fornasa*) that had the shape of truncated cones and an opening in the middle of their roofs. Such a shape of lime kilns was described in the documents, and is partially preserved on certain Croatian islands (Šolta, for example). This cartographic confirmation attests that the shape of lime kilns has not changed (significantly) over the course of many centuries.

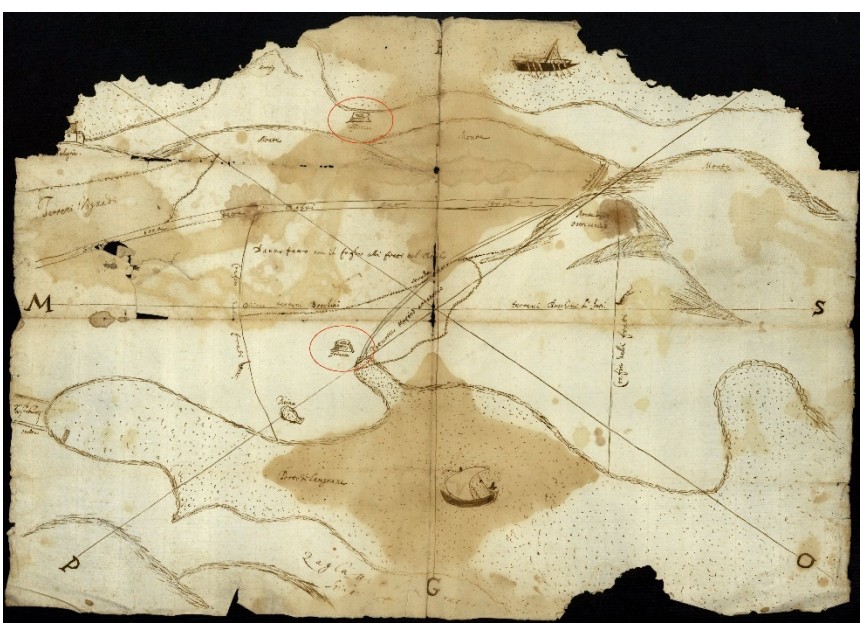

**Figure 7.** Drawing of a lime kiln as truncated cones with an opening in the middle of the roof (red ellipses) in a cadastral-topographic sketch of Kali (Ugljan Island), 1736.

On a map of Lukoran (Ugljan Island) made by the surveyor Bartolo Riuiera on 3 May 1736 (Figure 8), which is an attachment to the collection of documents *Per il Reverend: Arciprete Grisogono Al Laudo sopra Terre poste a Lucorano 1737* [76]), alongside the cove of Južnja Frnaža, are drawings of two lime kilns marked *Calchera* and *Fornace*, and another Calchera was drawn somewhat further away from the shore. Next to *Calchera* on the shore is written the toponym *Camegnac* (from the Croatian word *kamen*, "stone"), which indicates that there was a quarry used for the extraction of stone for the purposes of lime production in lime kilns nearby.

In 1782, Conte Lorenzo Licini Rubčić drew up a map in Zadar showing the demarcation between settlements on the north-western part of the island of Pašman *Dissegno che demonstra la linea di Confine delle Ville di Sdrelaz con Bagno e con Dobropogliana* [77]. On that map (Figure 9) are drawings of 22 lime kilns, 18 of which were built along the shores of Srednji kanal (a part of Zadar's local waters), and four of them were erected slightly farther from the shore. Each lime kiln (*calchera*) was marked with a red circle and a number, and an index of symbols contained the name of each location and number indicating where each lime kiln was built. For some of these, a note indicates they were old (*Cal. Vechia*) or new (*Calchera nuova*), which suggests the existence of continuous lime production and differentiation of lime kilns according to their antiquity.

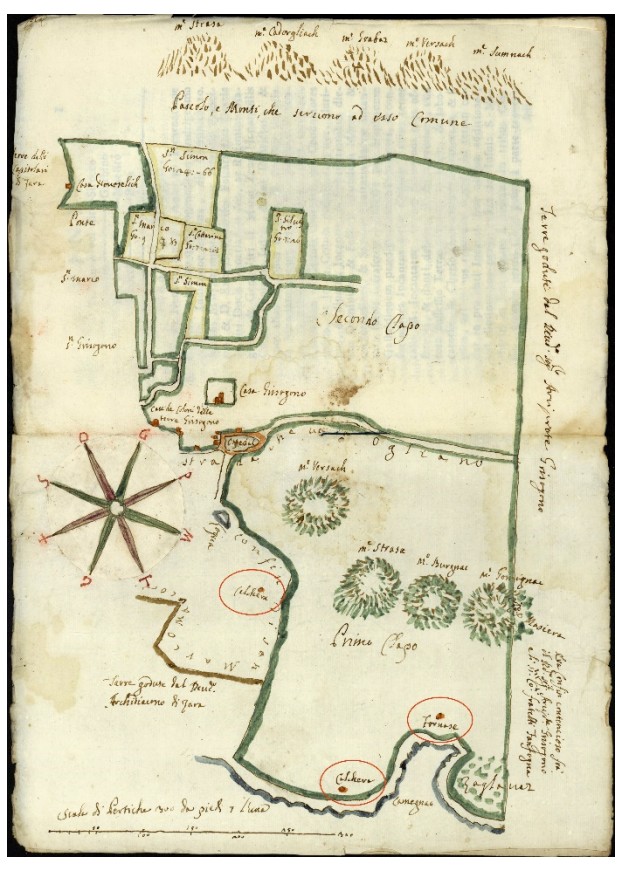

**Figure 8.** Cartographic depiction (red ellipses) of lime kilns in Lukoran on the island of Ugljan, 1736.

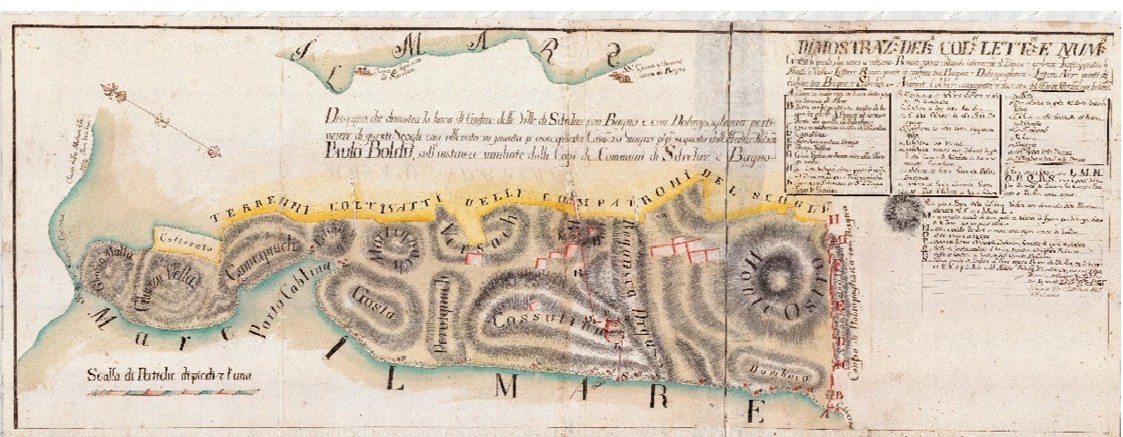

**Figure 9.** Licini Rubčić's map of the north-western part of the island of Pašman with depiction of lime kilns, 1782. The mapmaker marked each lime kiln with a circle and an identifying number.

### 4.4. Lime Production's Mark on the Linguistic Landscape

The linguistic landscape is substantially connected to the geographical features of an area. The key role in this regard is played by geographical names that, in the shortest possible manner, identify and differentiate various geographical facilities, which enables orientation regarding the determination of spatial relations and the execution of various activities. The naming of geographical facilities was often connected to the visual aspect and function of the named facility. Therefore, toponyms are often directly connected with appellatives used to differentiate various relief, hydrographic, biological, social, and economic elements in a given area—in relation to its specific lexis. The process of

toponymization (transforming appellatives into a geographical names) encompasses basic facilities used for lime production. In regard to this activity, on the northern Dalmatian islands, Croatian Slavic toponymy prevails, in a variant corresponding to the Čakavian dialect of the Croatian language. Therefore, lime kilns were most often named after a variant corresponding to the names of the facilities used for lime production (*japnenica*, *japjenica*, *japlenica*, "lime kiln"). Also used (although very rarely) are the Croaticized Roman words *klačina* (related to the Italian word *calchera*, "lime kiln") and *frnaža* (related to the Italian word *fornace*, "kiln/furnace"). For the sake of comparison, in other areas where lime production was an important economic activity, sites with lime kilns were given special names. In comparison, such toponyms (Calcinare, Bianchetta, Fornace) were recorded in Liguria, Italy [7], and due to the linguistic contact between Italian- and Croatian-speaking areas, these are similar to those recorded on the islands around Zadar and Šibenik.

During the exploration of insular toponymy, it was established that lime kilns were frequently named without any specific attribution (in terms of size, shape, ownership), which clearly indicates that, to the users of insular space, the mere existence of lime kilns was a sufficient realization differentiating the lime kiln location from the neighboring area. Only the locations with a greater number of lime kilns required more specific identification. Therefore, in such situations, adjectives were added to the basic name (e.g., *zmorašnja* "northwestern", *južnja* "southeastern", and other words with meanings such as "small" or "big").

The greatest number of preserved lime-kiln names refers to such facilities found along the shore. This is due to the fact that the vast majority of lime kilns were situated on the shore (for easier transportation of lime). In addition, taking into consideration the fact that not all lime kilns were named, certain lime kilns on the shore served as important landmarks for various usages of the coastal strip, especially during the fishing season; that is, positions (*pošte*) suitable for fishing due to habitats of good quality fish and characteristics of the sea bed (on which fishing equipment was not destroyed or damaged) were easy to identify in space by means of landmarks positioned on the shore. Lime kilns were among these landmarks on the sections of coast with uniform relief forms.

One geographical name does not testify to lime production but to the usage of the construction material itself. This is Japnjača on the island of Pag. This toponym refers to a section of an ancient Roman aqueduct on whose surface a layer of lime is clearly visible.

The quantity of toponyms connected to the production of lime generally refers to the intensity of that activity, which is clearly indicated by numerous toponyms on the islands of Ugljan, Pašman, and Pag, on whose territory the greatest number of lime kilns existed (Table 2, Figure 10). However, given that today there are no such toponyms in the linguistic landscape of certain northern Dalmatian islands, we cannot automatically conclude that there were no lime kilns in that area. For example, toponymy field research conducted on the islands of Kornat and Murter, regarded as larger islands in that Croatian archipelago, did not confirm the existence of a single toponym that could be directly linked to the production of lime, even though there is, in fact, no justifiable reason to doubt the existence of lime kilns on those islands. Such toponyms also do not exist on smaller inhabited islands, such as Ist, Rivanj, Vrgada, Ošljak, Prvić, Zlarin, Kaprije, and Krapanj. Similarly, on Dugi otok, which—apart from Pag—has the greatest land area of all the northern Dalmatian islands, toponyms connected with lime kilns were preserved only on the farthest south-eastern part of the island, even though similar social and economic developments occurred on the entire island over the course of many centuries. Moreover, field observations have confirmed the existence of remnants of lime kilns in those areas of Dugi otok where there were no toponyms that directly refer to such facilities. Field research carried out on the island of Škarda showed that there were approximately fifteen lime kilns, with only two of them having been given a name. One of these was named Belejska vapnenica ("Beli lime kiln") after the village of Beli on the island of Cres in Kvarner. According to local islanders, on Škarda there were lime kilns owned by the inhabitants of Škarda and Ist, in addition to those owned by the inhabitants of

Zapuntel on the island of Molat, Rab on the island of Rab, Nerezine on the island of Lošinj, and some that were owned by inhabitants of Bari in today's Italy [78].

**Table 2.** Toponyms connected with lime production in today's linguistic landscape of the northern Dalmatian islands (the islands are listed from northwest to southeast) [79–87].

| Island | Toponyms |
| --- | --- |
| Premuda | Japnenica |
| Silba | Punta Japlenica |
| Olib | Japlenica |
| Pag | Furnaža (three localities with this name), Japnenica (four localities with this name), Punta Frnaže, Klačina |
| Maun | Japnenica |
| Vir | Japnenica |
| Škarda | Japlenica, Belejska Japlenica |
| Molat | Japlenica, Japnenica |
| Zverinac | Pod Japlenicu |
| Sestrunj | Japleniško, Malo Japleniško, Punta Japleniškoga |
| Ugljan | Japlenica (two localities with this name), Mala Japlenica, Vela Japlenica, Zmorašnja Frnaža, Južnja Frnaža, Japleniško |
| Pašman | Japlenica (four localities with this name), Japlenice (two localities with this name), Pod Japlenicu, Put Japlenice |
| Babac | Japlenica na Prosiki, Japlenica u Barinon, Japlenica u Burića, Japlenica u Borin |
| Gangaro | Japnjenica |
| Iž | Japleniško |
| Rava | Japnenica, Pod Japnenicu |
| Dugi otok | Japnenica (three localities with this name) |
| Sit | Japjenica |
| Žut | Japjenica |
| Levrnaka | Japjenice, Pod Japjenice |
| Lavsa | Japjenica |
| Lucmarinjak | Japjenica |
| Kurba | Japjenica, Pod Japjenicu |
| Tijat | Japjenica |
| Žirje | Japljenišće, Punta Japlenišća |

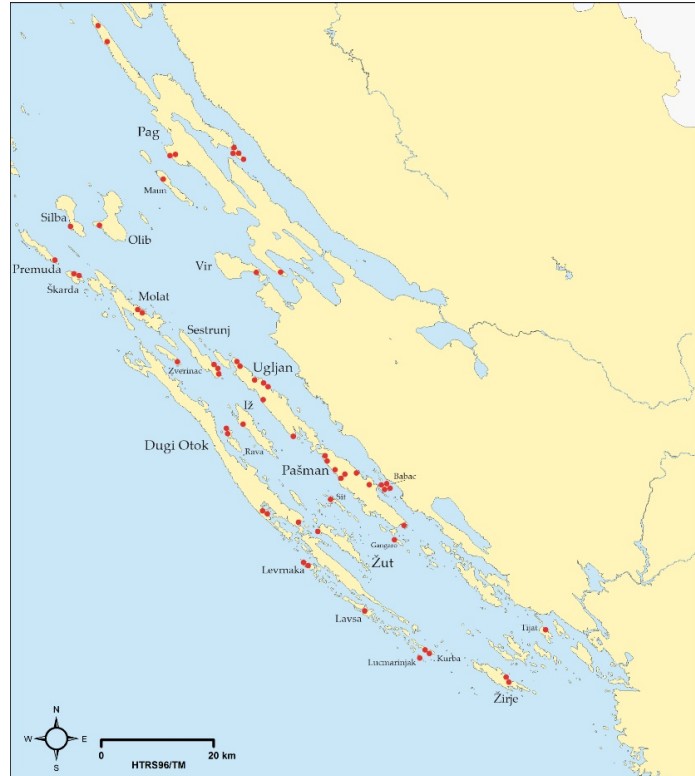

**Figure 10.** Geographical positions of toponyms in the northern Dalmatian islands referring to the production of lime.

Although there are no recorded toponyms related to lime production on certain islands, its existence has been confirmed in the archival sources. For example, in 1500 a contract was signed regarding the construction of a lime kiln at the locality of Luca Giuanschi on the island of Kaprije and, in 1506, a contract was signed regarding the construction of a lime kiln at the locality of Nosdra on the island of Krapanj [65]. Therefore, toponyms can be regarded as one means of proof of the footprint that lime production left in the (linguistic) landscape, but they are not fully reliable sources. It is not possible to consider them the only sources of spatial data that are conducive to making definitive conclusions.

## 5. Conclusions

The production of lime in the northern Dalmatian islands was an important form of the utilization of natural resources. This was substantially influenced by the prevailing carbonate bedrock of the islands in the Adriatic External Dinarides and the constant demand for lime, which was used in construction works and agriculture in the pre-industrial era. Lime kilns on the northern Dalmatian shorelines were located in the immediate vicinity of the main raw material (Upper Cretaceous ore-bearing limestone, Eocene Foraminifera limestone and, less frequently, Upper Cretaceous dolomite), the energent (firewood), and the coast because this facilitated the transport of lime to end consumers in island settlements and nearby cities on the mainland.

The available sources of relevant data have not enabled quantification in terms of determining concrete amounts of produced lime, or of the exploited stone, wood, and shrubs, which were the basic resources in such production. Moreover, apart from the exploitation of stone intended for lime production, significant quantities of stone were also used in the construction of residential and economic facilities, port infrastructure, and drystone walls for the purposes of estate demarcation and slope terracing to prevent soil erosion. In the same manner, trees and shrubs were used not only to fire lime kilns, but also for heating in residential buildings, in addition to the construction of houses, furniture, agricultural, and fishing equipment, and for catching pelagic fish at night. Due to these factors, it is not possible to make definitive assessments about the influence of lime production on changes in the insular environment and landscape. Nonetheless, by means of field and archival research, many facts have been determined that undoubtedly confirm that centuries of human lime production left a substantial footprint in the karst landscape. This is primarily manifested in anthropogenic forms of insular relief (lime kilns, smaller quarries, accumulations of rock) and in degraded forms of Mediterranean vegetation. The remnants of lime kilns on the North Dalmatian islands are small but important technofossils. These are piles of burnt stones and rubble that were formed in places where above-ground lime kilns were regularly built, with rock outcroppings in their foundations, on slightly sloping terrain. Their original form can be assumed based on both descriptions in historical documents and drawings on maps. These archival sources indicate similar forms of lime kilns as those preserved on the central Dalmatian island of Šolta, where traditional lime production was practiced until the mid-twentieth century.

In accordance with the deagrarianization and depopulation of the islands, the process of succession of original vegetation (a community of holm oaks) and the spread of allochthonous species, among which the Aleppo pine is the most common, is becoming increasingly intensive. These processes contribute to the reforestation of the islands, but also to the gradual disappearance of traces of the former lime production.

Given the importance of lime kilns in this island region, and the fact that they were erected in the immediate vicinity of the coast, many of them have become a part of the linguistic landscape. Under these circumstances of gradual disappearance of material traces, geographical names largely reflect the spatial distribution and intensity of the former activity of lime production. Numerous archival confirmations from the fourteenth century represent a lasting documentation of the exploitation of stone, the construction of lime kilns, and the cutting of timber for the production of lime on the islands for local and

regional needs, and these are supplemented by depictions of lime kilns on maps from the sixteenth century.

**Author Contributions:** Conceptualization, J.F. and K.J.; methodology, J.F.; formal analysis, J.F. and K.J.; investigation, J.F. and K.J.; resources, J.F. (old maps) and K.J. (archival records); data curation, J.F. and K.J.; writing—original draft preparation, J.F. and K.J.; writing—review and editing, J.F. and K.J.; visualization, J.F. and K.J.; supervision, J.F. and K.J. All authors have read and agreed to the published version of the manuscript.

**Funding:** The authors carried out this study as part of their regular scientific research at the University of Zadar.

**Institutional Review Board Statement:** Not applicable.

**Informed Consent Statement:** Not applicable.

**Data Availability Statement:** Data are contained within the article and in archival records that we cited in article.

**Acknowledgments:** We would like to thank Jadran Kale from the Department of Ethnology and Anthropology at the University of Zadar, who gave us access to his photographs of lime kilns on islands of Žirje, Orut, and Levrnaka.

**Conflicts of Interest:** The authors declare no conflict of interest.

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
