# Peer review of "Human Footprints in the Karst Landscape: The Influence of Lime Production on the Landscape of the Northern Dalmatian Islands (Croatia)"

_geosciences, doi:10.3390/geosciences11080303_

Round 1

Reviewer 1 Report

Human Footprint in the Karst Landscape: The Influence of Lime Production on Changes in the Landscape of the Northern Dalmatian Islands

Geosciences – Review Report Form

General comments

The paper deals with interesting topic. Anyway, the reader would expect to get some more information about changes in the landscape as it is promised in the title. Instead of this, we get detailed overview on history of lime production in the Northern Dalmatian islands, but only a few data about changes in the environment due to this past economic activity. There is nothing wrong in this, but, please, consider to change the title of the article.

To strengthen the general scientific importance of the results section a comparison with similar studies across the world is missing. I assume that lime production throughout the world was an important economic activity in the past, but probably differently organized, using different types of kilns, material etc. and a comparison (using literature) would contribute to the overall quality of the paper. You mention different methods of lime production in the Introduction section, but some discussion is missing in Results section. Especially in the Mediterranean region, lime production was important and readers would appreciate to get some information about similarities, differences in lime production in Mediterranean region in the past.

I miss a more detailed spatial analysis of the lime kilns and their influence on the landscape. Just showing locations in somewhat not enough.  The paper was submitted to the journal Geosciences and one would expect to get some quantitative analysis about material used to produce lime (types of limestone, types of dolomite), geological structure of the rock material used (fractured materials, thin-bedded layers etc.), micro locations of lime kilns, locations of quarries, number of workers involved in the business etc. It would be nice to get some insight into location factors for locations of lime kilns; was it the closeness to market, needs, available material, available energy resources (wood), market policy, ownership etc.

Here are some further recommendations for the authors following the sections of the paper. Please, consider also corrections, marked text and comments in the pdf file.

  1. Introduction
  • I suggest to mention that production of lime is just one of the possible human impacts, type of landscape degradation of karst environment. For example, especially in modern times, many parts of the Dinaric karst dolines have been extensively filled up for the purposes of road, railway or building construction. You can find some information regarding this in papers with DOI:
    • https://doi.org/10.5937/GeoPan1004109C
    • https://doi.org/10.1016/j.scitotenv.2013.01.002
    • https://doi.org/10.1007/s12371-021-00544-0
  • Put some implications on work, previously done by other researchers, dealing with problems of lime production in connection to changes in karst environment.
  • See comments in the paper.
  1. Materials and Methods
  • Qualitative approach was used, some quantitative analysis is missing as already mentioned in the General comments section.
  • See comments in the paper.

 Results

  • Restructure this very long section in subsections, please.
  • Could you please put a sketch of a typical lime kiln of this region. Were they the type of a single or multiple use?
  • To strengthen the novelty and the overall value of the results of the research and their general scientific importance a comparison with similar studies across the world is missing. Otherwise, the results could be considered only as a case study results.

References

  • I suggest putting some additional references into Introduction and Results section.

Date of this review: 8 June 2021

Author Response

Dear Reviewer,

we received and carefully studied the reviews as well as the suggestions of the academic editors. Following specific comments and advice, for which we are especially grateful, we approached the refinement of our manuscript with the aim of improving its quality.

We did the following:

  1. We have changed the title in accordance with the recommendation that there are not enough reliable data on the basis of which it is possible to conclude what and how many changes in the landscape are in question; the impact of lime production is undeniable but it is difficult to quantify the changes and track their chronological sequence
  2. We have added a chapter in which the Study area is briefly described
  3. We have supplemented the chapter on methodology so that it is clearer what exactly we did, what our main data sources were and what methods we applied
  4. The chapter on results was further elaborated and divided into additional subchapters
  5. We have added an overview of different forms of human influence on the formation of karst landscape on the northern Dalmatian islands, using the results of landscape research in the Dinaric karst of other authors
  6. We compared the impact of lime production on the landscape of the northern Dalmatian islands with similar appearance in other Mediterranean countries; this comparison is appropriate because of the similarities in lithological composition, climate, vegetation and, consequently, landscape as well as the analogy in the methods of lime production
  7. In accordance with what we have done under points 5 and 6, we have supplemented the introduction and the results with relevant references.
  8. We have added references to figures and tables in the text.
  9. We have better explained the influence of location factors on lime production on the North Dalmatian islands.
  10. We have shortened the text in the introductory part of the subchapter on the depictions of limestones on old maps
  11. We wrote in more detail about the types, geological age and characteristics of limestone and dolomite which are the main raw material for lime production and added a clip of the Geological Map of the Republic of Croatia, original scale 1: 300,000
  12. We supplemented the conclusion with clearly highlighted forms of the impact of lime production on the landscape of the northern Dalmatian islands
  13. In the absence of material remains of limestone that would clearly indicate the shape of lime kilns on the northern Dalmatian islands, we have added photographs of lime kilns from the island of Solta (taken in the middle of the 20th century) located in the nearby Central Dalmatian archipelago
  14. On the old maps, we marked the locations with lime kilns with red ellipses. We did not do this on Licini Rubcic's map because this author from the 18th century marked the lime kilns with red circles in many places (we referred to this in the text below the title of that map)
  15. We have made minor corrections and additions in all places where they are suggested into the manuscript, relate to the text and images.

We have marked all changes and additions in the text in colour (blue – replaced, red – added and green – shortened) so that they can be noticed more easily.

Josip Faričić and Kristijan Juran

Reviewer 2 Report

Dear authors,

thank you for your article which, broadly speaking, can also be interesting. However, in my opinion, there are a number of fundamental sections that are missing to make sure that this article can be accepted in a journal like Geosciences.

1) You speak of karst landscape but lack any reference and information about it; there is no geological map, no geomorphological map, no photos of sinkholes, karren, or any karst landforms. I would suggest inserting a "study area" paragraph in which these issues are addressed in detail.

2) There is a completely missing section of methodology that is fundamental to a scientific article nowadays. You cannot do a scientific article without every single topic being explained in the methodology in every detail.

3) There are no references in the text to the figures and tables that are fundamental to make the reader understand what they refer to.

4) The aim of this work, the innovations it highlights, and what it can bring to the scientific community are not well explained. In addition, a scientific article must also be repeatable by the community, but the absence of these fundamental parts does not mean that this article can be defined as a complete scientific article.

Ultimately, in my opinion, the article needs a great deal of work to make sure that it can be accepted.

I attach the pdf for minor revisions

Best regards

Author Response

(The authors gave the same response as above.)

Round 2

Reviewer 2 Report

Dear authors,

Thank you for listening to my suggestions and reviewing the article satisfactorily. I enclose some minor revisions.

Author Response

Dear Reviewer,

we have received suggestions for minor changes related to images (that you suggested) and text (that Academic Editors suggested). We are grateful for this additional effort that has been made to improve the quality of the final form of the manuscript.

In line with these proposals, we have done the following:

- we supplemented the geological map by inserting essential parts (compass rose, scale and legend)

- we have improved the figure captions with important facts so the figures (with captions) can be self-standing.

We also carefully checked the reference numbers throughout the text and made several minor terminological changes, including a substitution related to naming the geological time determinant (instead of ‘Tertiary’ we used the names Paleogene and Neogene).

We marked all changes in the latest version of the text in red and additionally marked them with a yellow marker to make them easier to see.

Best regards,

Josip Faricic and Kristijan Juran